# Quality of life and its associated factors in chronic kidney disease patients undergoing hemodialysis from a Peruvian city: A cross-sectional study

Dana Machaca-Choque[1], Guimel Palomino-Guerra[1], Javier Flores-Cohaila[2],
Edgar Parihuana-Travezaño[1], Alvaro Taype-Rondan[3,4], Sujey Gomez-Colque[5],
Cesar Copaja-Corzo[3,6]*

**1** Facultad de Ciencias de la Salud, Universidad Privada de Tacna, Tacna, Peru, **2** Facultad de Ciencias de la Salud, Universidad Científica del Sur, Lima, Peru, **3** Unidad de Investigación para la Generación y Síntesis de Evidencias en Salud, Universidad San Ignacio de Loyola, Lima, Peru, **4** EviSalud—Evidencias en Salud, Lima, Peru, **5** Facultad de Ciencias de la Salud, Universidad Nacional Jorge Basadre Grohmann, Tacna, Peru, **6** Servicio de Infectología, Hospital Nacional Edgardo Rebagliati Martins, EsSalud, Lima, Perúu

\* Csarcopaja@gmail.com

## Abstract

### Objective

To assess quality of life and explore its associated factors in a group of patients with chronic kidney disease (CKD) undergoing hemodialysis in Peru.

### Methodology

We conducted a cross-sectional analysis of patients with CKD treated at two medical centers in Tacna, Peru; between July and September 2023. We conducted a survey via telephone interviews with eligible patients using the Short Form 36 (SF 36) to assess their quality of life.

### Results

Of 257 patients with CKD undergoing hemodialysis, we successfully interviewed 207 (59.9% males, median age: 62 years, median time on hemodialysis: 3.5 years). In the context of the SF-36 assessment, the dimensions with the lowest scores were physical role (mean: 13.9), emotional role (32.2), and physical function (32.4). Regarding the SF-36 summary scores, the average scores were 42.2 in the mental health domain and 32.0 in the physical health domain. In the adjusted model, the physical health domain score was higher in males (β = 2.7) and those with economic self-sufficiency (β = 3.0) and lower in older adults (β = -2.5). The score in the mental health domain was higher in those with a higher level of education (β = 4.1), in those with economic self-sufficiency (β = 3.8), and in those receiving care at one of the centers included (β: 4.2).

**Data Availability Statement:** All relevant data are within the manuscript and its Supporting Information files.

**Funding:** This work received funding to cover the article processing charge (APC) by the following sources of support: Universidad San Ignacio de Loyola (to C.C.C.) and Universidad Científica del Sur (to J.F.C.). The funders had no role in the study design, data collection and analysis, decision to publish, or manuscript preparation.

**Competing interests:** The authors have declared that no competing interests exist.

## Conclusion

Quality of life was affected, particularly in the realms of physical and emotional well-being. Furthermore, both the physical and mental health domains tend to show lower scores among women, older individuals, those lacking economic self-sufficiency, individuals with lower educational levels, and those with comorbidities.

## Introduction

Hemodialysis is the treatment for kidney failure (KF) [1], and its main benefit is prolonging life. However, these life-years gained can be associated with low Health-Related Quality of Life (HRQoL) [2, 3]. Studies indicate that, when contrasted with individuals grappling with other chronic conditions like cancer or heart failure, patients with Chronic Kidney Disease (CKD) tend to experience a lower level of HRQoL [4, 5].

There is a nexus between HRQoL, morbidity and mortality in patients with CKD [6–8]. Owing to its significant impact, research on HRQoL in CKD has increased [9], and have found that some factors such as disease-related clinical manifestations, treatment side effects, and mental health issues such as anxiety, depression, and psychological distress, can negatively influence the HRQoL of patients with CKD [10, 11]. Gaining a deeper comprehension of these factors holds the key to formulating proactive interventions aimed at improving the HRQoL in these patients.

However, despite the existing literature on factors associated with HRQoL in patients with CKD undergoing hemodialysis, there is a lack of studies that describes this phenomenon in Latin America [2]. To address this gap, this study aimed to identify factors associated with HRQoL using the SF-36 questionnaire in hemodialysis patients with KF from two hemodialysis centers in Tacna, Peru.

## Methodology

### Study design and setting

We conducted a cross-sectional analytical study of patients treated at two hemodialysis centers in Tacna, Peru, between July 1st and September 30th, 2023. We adhered to the guidelines set forth by Strengthening the Reporting of Observational Studies in Epidemiology (STROBE) for reporting observational studies [12].

In the city of Tacna, Peru, there are two main centers that provide hemodialysis treatment to patients with health insurance:

- The Kidney Clinic, a private institution specializing in hemodialysis services, caters to patients affiliated with the Comprehensive Health Insurance (known as SIS in Spanish). Equipped with nine state-of-the-art hemodialysis machines, the clinic currently serves a roster of 132 registered patients [13]. Notably, the hospitals under the purview of the Peruvian Ministry of Health in Tacna refer patients in need of hemodialysis to the Kidney Clinic, as these hospitals lack the requisite infrastructure for providing this specialized care. The SIS, a health insurance program, is specifically designed to safeguard the well-being of Peruvians without alternative health coverage, with a particular emphasis on vulnerable populations grappling with poverty or extreme poverty [14].

- The Hemodialysis Unit of the Daniel Alcides Carrión III Hospital (DACH) in Tacna has 10 hemodialysis machines and 125 registered patients. The patients treated at this institution are affiliated with the Social Health Insurance (known as EsSalud in Spanish), a coverage tailored for employed individuals [15].

## Population

We included adult patients ($\geq$ 18 years old) diagnosed with KF undergoing hemodialysis and receiving care at one of two hemodialysis centers under study in Tacna, Peru. The diagnosis of KF was performed by a nephrology specialist according to the KDIGO guidelines [1], where KF or stage V CKD is defined as glomerular filtration rate $<$ 15 ml/min/1.73 m$^2$ [16]. We excluded patients hospitalized during their evaluation, those who did not respond to phone calls, those with insufficient information in their medical records and those who did not wish to participate in the study.

## Procedures

To collect data for our study, we requested access to the patient database of both the Hemodialysis Unit at the DACH and the Kidney Clinic in July 1st 2023. After obtaining the necessary permissions and the list of patients treated at these centers, between July 5th and July 10th 2023, we identified patients who met the inclusion criteria and verified their contact information and phone numbers.

Two researchers (D.M.C. and G.P.G.) underwent two training sessions in interviews and data collection, each lasting 45 minutes. Between July 10th and August 31st 2023, we conducted interviews with the patients via telephone as follows: during the call the researchers described the informed consent, explain the research objectives, and the risks and benefits of participating. After explaining this, they requested a verbal response from the patients regarding their willingness to participate, and this was documented in writing in the survey. If the patient agreed to participate in the study, data collection from the questions was initiated. If the patient had any doubts during the interview, the researchers addressed the concern. If the patient did not wish to provide an answer, the researchers omitted the question and moved on to the next one.

Between August 31st and September 30th 2023, the researchers collected data from the medical records. To do this, they identified the number of medical records and gathered information on the variables of interest. A third researcher (C.C.C.) conducted quality control on the collected information. If any discrepancies were found, he searched for information in the medical records and made the necessary corrections.

## Questionnaire and variables

The data collection form was divided into four sections as follows: (1) sociodemographic characteristics (7 questions), (2) clinical characteristics (13 questions), (3) HRQoL (36 questions), and (4) other scales to assess aspects of mental health (**S1 File**).

To assess HRQoL, we used the SF-36 health questionnaire. This instrument is designed to evaluate health concepts that are relevant across age groups, diseases, and treatments in adults [17]. SF-36 comprises eight domains: physical functioning (10 items), role limitations due to physical health (4 items), bodily pain (2 items), general health (5 items), vitality (4 items), social functioning (2 items), role limitations due to emotional health (3 items), and mental health (5 items). Within each dimension, the respondent receives a score from 0 to 100. A higher score correlates with better HRQoL [6]. Additionally, scores were calculated for two

summary components: Physical Component Summary and Mental Component Summary. Their calculation was performed following the scoring manual for the Spanish version of the SF-36 Health Survey [18].

In our study, we used the Spanish version of the SF-36, validated in an adult population (≥18 years) across multiple contexts [19]. It was reported to have an internal consistency coefficient or combined Cronbach's alpha estimate greater than 0.7 on all scales, being greater than to 0.9 on the physical functioning, physical health, and emotional health scale in general population studies [19].

## Statistical analysis

All analyses were conducted using statistical software Stata v17 (StataCorp LLC). To describe participant characteristics, we used absolute and relative frequencies; as well as means ± standard deviations (m ± sd) for numerical variables.

To assess the factors associated with HRQoL, we considered the Physical and Mental Component Summary scores of the SF-36 as numeric outcomes. We used linear regression models to calculate the coefficients (β) and their respective 95% confidence intervals (95% CI). For this purpose, we initially conducted an unadjusted linear regression analysis for each independent variable, and those with statistically significant associations ($p < 0.05$) were included into the adjusted regression model.

## Ethics

The ethical guidelines outlined in the Declaration of Helsinki were followed. The study was reviewed and approved by the ethics committee of the Faculty of Health Sciences at the Private University of Tacna (code: FACSA-CEI/032-06-2023). The patients provided their oral informed consent to participate in this study to the researchers during the interview, and this was documented in writing in the survey.

## Results

### Population characteristics

We identified 257 patients diagnosed with KF undergoing hemodialysis at the evaluated centers between July 1st and September 31st, 2023. Of these, 140 were treated at the Hemodialysis Unit of DACH in Tacna and 117 at the Kidney Clinic. Fifty patients were excluded (23 did not answer the phone call, 13 had incomplete clinical data, 11 were hospitalized, and three declined to participate), resulting in the final inclusion of 207 patients in the present study (**Fig 1**).

The median age was 62 years, 59.9% were male, the majority (53.1%) had received hemodialysis through Hemodialysis Unit of DACH, 81.6% had only completed primary education, and 75.4% were financially dependent on their families.

Regarding clinical characteristics, most patients had one to two comorbidities, the most common being hypertension (71.5%), diabetes mellitus (44.4%), and total blindness (24.6%). Most patients (83.6%) received hemodialysis three times a week, and each hemodialysis session lasted 3 hours for 53.1% of patients. The remaining 46.9% of patients underwent hemodialysis sessions lasting three and a half hours. The median number of years patients had been on hemodialysis was 3.5 years (range:1.5–6.5) (**Table 1**).

### Quality of life score

The dimensions of the SF-36 questionnaire with the highest mean scores were mental health (mean 61, sd ±16.7), social functioning (59.2 ± 21.2), and vitality (mean 53.4, sd ±8.4). Among

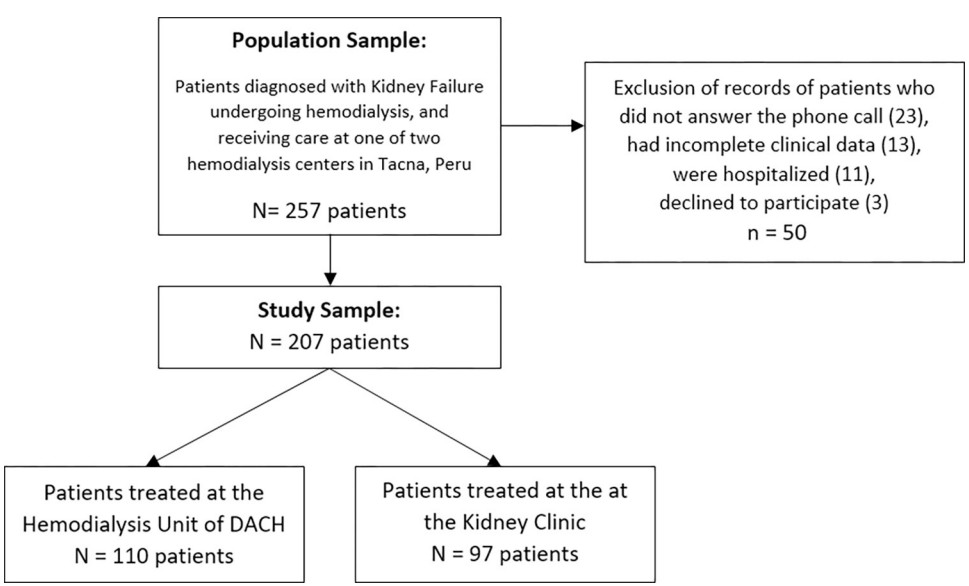

**Fig 1. Flowchart detailing the process of sample selection (N = 207).**

the patients from the Hemodialysis Unit of DACH, the dimensions with the highest scores were mental health (mean 65.2, sd ±16.8) and social functioning (mean 59.1, sd ±20.5). On the other hand, among the patients from the Kidney Clinic hemodialysis unit, the dimensions

**Table 1. Population characteristics and distribution by hemodialysis center.**

| Characteristics | Total (n = 207) n (%) | Hemodialysis Unit of DACH (n = 110) n (%) | Kidney Clinic (n = 97) n (%) |
|---|---|---|---|
| Sex: male | 124 (59.9) | 62 (56.4) | 47 (48.5) |
| Older age: ≥60 | 124 (59.9) | 72 (65.5) | 52 (53.6) |
| Marital status: married | 144 (69.6) | 82 (74.6) | 62 (63.9) |
| Education: only primary education | 169 (81.6) | 80 (72.7) | 89 (91.8) |
| Financially dependent on the family | 156 (75.4) | 70 (63.6) | 86 (88.7) |
| Urban residency | 196 (94.7) | 109 (99.1) | 87 (89.7) |
| Comorbidities | | | |
| Hypertension | 148 (71.5) | 72 (65.5) | 76 (78.4) |
| Diabetes mellitus | 92 (44.4) | 51 (46.4) | 41 (42.3) |
| Total blindness | 51 (24.6) | 24 (21.8) | 27 (27.8) |
| Autoimmune disease | 15 (7.3) | 9 (8.2) | 6 (6.2) |
| Peripheral vascular disease | 14 (6.8) | 12 (10.9) | 2 (2.1) |
| Congestive heart failure | 8 (3.9) | 6 (5.5) | 2 (2.1) |
| Pulmonary tuberculosis | 5 (2.4) | 5 (4.6) | 0 (0.0) |
| Number of comorbidities | | | |
| None | 19 (9.2) | 10 (9.1) | 9 (9.3) |
| One to two | 146 (70.5) | 76 (69.1) | 70 (72.2) |
| Three to four | 42 (20.3) | 24 (21.8) | 18 (18.6) |
| Time on hemodialysis | | | |
| Less than three years | 85 (41.1) | 42 (38.2) | 43 (44.3) |
| Three years to less than six years | 64 (30.9) | 32 (29.1) | 32 (33.0) |
| Six years or more | 58 (28.0) | 36 (32.7) | 22 (22.7) |

**Table 2. Distribution of scores for each SF-36 dimension and distribution by hemodialysis center.**

| Dimensions | Total score (m ± sd) | Score of patients from the Hemodialysis Unit of DACH (m ± sd) | Score of patients from the Kidney Clinic (m ± sd) |
|---|---|---|---|
| SF-36 dimensions | | | |
| Mental health | 61.0 ± 16.7 | 65.2 ± 16.8 | 56.1 ± 15.2 |
| Social function | 59.2 ± 21.2 | 59.1 ± 20.5 | 59.3 ± 22.0 |
| Vitality | 53.4 ± 18.4 | 58.0 ± 18.2 | 48.2 ± 17.2 |
| Corporal pain | 50.9 ± 23.8 | 50.8 ± 23.4 | 50.9 ± 24.4 |
| General health | 37.7 ± 18.2 | 39.9 ± 19.3 | 35.2 ± 16.6 |
| Physical function | 32.4 ± 29.7 | 33.1 ± 29.4 | 31.6 ± 30.1 |
| Emotional role | 32.2 ± 44.2 | 40.6 ± 47.0 | 22.7 ± 39.0 |
| Physical role | 13.9 ± 26.9 | 16.6 ± 30.5 | 10.8 ± 21.9 |
| SF-36 summary scores | | | |
| Mental component summary | 42.2 ± 11.0 | 45.0 ± 11.1 | 39.1 ± 10.0 |
| Physical component summary | 32.0 ± 8.4 | 31.5 ± 8.8 | 32.5 ± 7.9 |

m: mean; sd: standard deviation. HRQoL score has a score between 0 (the worst health state for that HRQoL dimension) and 100 (the best health state for that HRQoL dimension).

with the highest scores were social functioning (mean 59.3, sd ±22) and mental health (mean 56.1, sd ±16.2). Additionally, Mental Component Summary had the highest mean score. (mean 42.2, sd ±11) (**Table 2**).

## Factors associated with quality of life

In the adjusted model, we found that the physical health score of SF-36 was higher in males (β: 2.7, 95% CI: 0.5 to 4.8) compared to females and in those who were financially self-sufficient (β: 3.0, 95% CI: 0.3 to 5.7) compared to those who relied solely on family financial support. On the other hand, the physical health score was lower in older adults (β: -2.5, 95% CI: -4.8 to -0.1) compared to adults aged 20 to 59, and individuals with hypertension and diabetes also had lower scores (β: -4.8, 95% CI: -8.1 to -1.5) compared to those without these comorbidities (**Table 3**).

Regarding the mental health score of SF-36, patients treated at the Kidney Clinic had lower scores (β: -4.22, 95% CI: -7.3 to -1.2) compared to DACH patients. On the other hand, those who were financially self-sufficient had higher scores (β: 3.80, 95% CI: 0.3 to 7.3) compared to those who relied solely on family financial support, and individuals with higher educational levels also had higher scores (β: 4.07, 95% CI: 0.2 to 7.9) compared to those with only primary education (**Table 3**).

## Discussion

In this study, we described the factors associated with HRQoL using the SF-36 questionnaire in 207 hemodialysis patients with CKD. The SF-36 dimensions with the lowest scores were physical role, emotional role, and physical function. In the adjusted model, the physical health domain score was higher in males and in those with economic self-sufficiency, and lower in older adults. The score in the mental health domain was higher in those with a higher level of education, in those with economic self-sufficiency, and in those receiving care at one of the centers included.

Our results align with previous studies that have reported an association between male gender and better self-perceived HRQoL [20]. Has been reported that gender has an important

**Table 3.** Factors associated with the physical and mental health summary scores of the SF-36 questionnaire (n = 207).

| Characteristics | Outcome: SF-36 Physical health score | | | Outcome: SF-36 Mental health score | | |
|---|---|---|---|---|---|---|
| | m ± sd | Crude β (95% CI) | Adjusted β (95% CI)[b] | m ± sd | Crude β (95% CI) | Adjusted β (95% CI)[c] |
| Gender | | | | | | |
| Female | 30.3 ± 7.4 | Ref | **Ref** | 40.9 ± 10.3 | Ref | - |
| Male | 33.5 ± 8.9 | 3.2 (0.9 to 5.4) | **2.7 (0.5 to 4.8)** | 43.4 ± 11.5 | 2.5 (-0.5 to 5.5) | - |
| Age[a] | | | | | | |
| Adult (20 to 59) | 34.5 ± 7.9 | Ref | **Ref** | 43.9 ± 11.5 | Ref | - |
| Older adult (≥60) | 30.3 ± 8.3 | -4.2 (-6.5 to -1.9) | **-2.5 (-4.8 to -0.1)** | 41.1 ± 10.5 | -2.8 (-5.9 to 0.2) | - |
| Healthcare facility | | | | | | |
| Hemodialysis Unit of DACH (Three hours of HH) | 31.5 ± 8.8 | Ref | - | 45.0 ± 11.1 | Ref | **Ref** |
| Kidney clinic (Three and a half hours of HH) | 32.5 ± 7.9 | 0.9 (-1.4 to 3.2) | - | 39.1 ± 10.0 | -6.0(-8.9 to -3.0) | **-4.3 (-7.3 to -1.2)** |
| Marital status | | | | | | |
| Without partner | 31.7 ± 8.1 | Ref | - | 41.6 ± 10.8 | Ref | - |
| With partner | 32.1 ± 8.5 | 0.5 (-2.0 to 3.0) | - | 42.5 ± 11.1 | 1.0 (-2.3 to 4.2) | - |
| Level of education | | | | | | |
| Without studies | 29.5 ± 8.9 | Ref | - | 39.7 ± 10.9 | Ref | Ref |
| With primary studies | 32.0 ± 8.3 | 2.6 (-2.2 to 7.3) | - | 41.2 ± 10.9 | 1.5 (-4.6 to 7.6) | -0.5 (-6.5 to 5.5) |
| With higher technical or university studies | 32.6 ± 8.4 | 3.2 (-2.2 to 8.5) | - | 47.5 ± 10.1 | 7.8 (1.0 to 14.6) | 3.6 (-3.3 to 10.5) |
| Financial support | | | | | | |
| Family support | 30.4 ± 8.1 | Ref | **Ref** | 41.0 ± 10.3 | Ref | **Ref** |
| Self-sufficiency | 35.5 ± 8.0 | 5.6 (3.1 to 8.2) | **3.0 (0.3 to 5.7)** | 44.9 ± 12.1 | 6.1 (2.7 to 9.5) | **3.8 (0.3 to 7.3)** |
| Residency | | | | | | |
| Urban | 32.1 ± 8.5 | Ref | - | 42.3 ± 11.0 | Ref | - |
| Rural | 29.9 ± 6.5 | -2.2 (-7.3 to 2.9) | - | 41.7 ± 11.8 | -0.6 (-7.3 to 6.2) | - |
| Hypertension | | | | | | |
| No | 35.0 ± 8.3 | Ref | - | 42.0 ± 12.0 | Ref | - |
| Yes | 30.8 ± 8.1 | -4.2 (-6.7 to -1.7) | - | 42.3 ± 10.6 | 0.3 (-3.1 to 3.6) | - |
| Diabetes mellitus | | | | | | |
| No | 33.4 ± 8.5 | Ref | - | 41.7 ± 10.8 | Ref | - |
| Yes | 30.2 ± 7.9 | -3.2 (-5.4 to -0.9) | - | 43.0 ± 11.2 | 1.3 (-1.8 to 4.3) | - |
| Comorbidity | | | | | | |
| No diabetes mellitus or hypertension | 35.9 ± 8.0 | Ref | Ref | 42.5 ± 13.0 | Ref | - |
| Hypertension or diabetes mellitus | 32.6 ± 8.5 | -3.4 (-6.5 to -0.2) | -2.2 (-5.2 to 0.8) | 41.4 ± 9.9 | -1.1 (-5.4 to 3.1) | - |
| Hypertension and diabetes mellitus | 29.0 ± 7.3 | -6.9 (-10.2 to -3.6) | **-4.8 (-8.1 to -1.5)** | 43.5 ± 11.5 | 1.0 (-3.5 to 5.5) | - |
| Congestive heart failure | | | | | | |
| No | 32.0 ± 8.2 | Ref | - | 42.2 ± 11.0 | Ref | - |
| Yes | 31.8 ± 13.4 | -0.2 (-6.2 to 5.8) | - | 42.8 ± 10.9 | 0.6 (-7.2 to 8.4) | - |
| Peripheral vascular disease | | | | | | |
| No | 32.2 ± 8.3 | Ref | - | 41.9 ± 10.9 | Ref | - |
| Yes | 28.7 ± 8.3 | -3.6 (-8.1 to 1.0) | - | 47.4 ± 10.6 | 5.5 (-0.5 to 11.4) | - |
| Pulmonary tuberculosis | | | | | | |
| No | 32.1 ± 8.4 | Ref | - | 42.3 ± 11.1 | Ref | - |
| Yes | 28.3 ± 5.6 | -3.8 (-11.3 to 3.7) | - | 41.7 ± 7.3 | -0.6 (-10.4 to 9.3) | - |
| Total blindness | | | | | | |
| No | 32.7 ± 8.3 | Ref | Ref | 42.6 ± 11.3 | Ref | - |
| Yes | 29.8 ± 8.2 | -2.9 (-5.5 to -0.2) | -1.3 (-3.8 to 1.3) | 41.3 ± 9.9 | -1.3 (-4.8 to 2.2) | - |
| Autoimmune disease | | | | | | |

*(Continued)*

**Table 3.** (Continued)

| Characteristics | Outcome: SF-36 Physical health score | | | Outcome: SF-36 Mental health score | | |
|---|---|---|---|---|---|---|
| | m ± sd | Crude β (95% CI) | Adjusted β (95% CI)[b] | m ± sd | Crude β (95% CI) | Adjusted β (95% CI)[c] |
| No | 31.9 ± 8.4 | Ref | - | 42.5 ± 10.9 | Ref | - |
| Yes | 32.4 ± 7.7 | 0.5 (-3.9 to 4.9) | - | 39.4 ± 12.5 | -3.0 (-8.8 to 2.8) | - |
| Number of comorbidities | | | | | | |
| None | 33.4 ± 8.7 | Ref | - | 45.7 ± 13.8 | Ref | - |
| One to two | 32.8 ± 8.3 | -0.6 (-4.6 to 3.3) | - | 41.6 ± 10.6 | -4.2 (-9.4 to 1.1) | - |
| Three to four | 28.5 ± 7.7 | -5.0 (-9.4 to -0.5) | - | 43.0 ± 10.7 | -2.7 (-8.7 to 3.3) | - |
| Frequency of hemodialysis per week | | | | | | |
| Twice a week | 31.4 ± 8.7 | Ref | - | 43.2 ± 11.5 | Ref | - |
| Three times a week | 32.1 ± 8.3 | 0.7 (-2.4 to 3.8) | - | 42.1 ± 10.9 | -1.1 (-5.2 to 2.9) | - |
| Frequency of hemodialysis per month | | | | | | |
| Four to eleven | 32.1 ± 7.8 | Ref | - | 41.6 ± 10.2 | Ref | - |
| Twelve to thirteen | 31.7 ± 9.0 | -0.4 (-3.3 to 2.6) | - | 42.4 ± 11.8 | 13.0 (0.7 to -3.1) | - |
| Fourteen plus | 32.2 ± 8.2 | 0.1 (-2.8 to 3.1) | - | 42.6 ± 10.7 | 14.0 (0.6 to -2.8) | - |
| Time on haemodialysis in years[a] | | | | | | |
| One to two years 11 months 29 days | 32.8 ± 8.3 | Ref | - | 43.9 ± 11.0 | Ref | - |
| Three to five years 11 months 29 days | 31.7 ± 8.9 | -1.1 (-3.8 to 1.6) | - | 40.6 ± 11.2 | -3.3 (-6.9 to 0.3) | - |
| Six or more year | 31.1 ± 7.9 | -1.6 (-4.5 to 1.2) | - | 41.7 ± 10.5 | -2.2 (-5.9 to 1.4) | - |

m: mean; sd: standard deviation, HH: Hours of hemodialysis

[a]Median (interquartile range)

[b]The variables that entered the multivariate model were gender, age, financial support, comorbidity, and blindness.

[c]The variables that entered the multivariate model were healthcare facility, level of education and financial support.

effect on the determinants and consequences of health and illness and may be explained by health inequities, because women in developing countries tend to have less social support, lower income, and lower healthcare utilization than men [20, 21].

In our study, we found lower physical health score in individuals with comorbidities like hypertension and diabetes, which is consistent with the findings of previous research [22–24]. We also found a lower physical health score in older adults. This finding is supported by previous studies, and can be explained by the reduction in strength, energy, and self-care capacity of elderly patients given that patients undergoing hemodialysis must follow strict diets and take medications frequently [25, 26].

In the psychological sphere, self-sufficiency, and higher educational attainment (technical or university education) were associated with better HRQoL, while being treated at the Kidney Clinic was associated with worse HRQoL. Interpreting these results, both fall within the realm of socioeconomic status, since higher educational levels are generally associated with higher income and these patients may have a better understanding of the course of the disease, the importance of treatment, and the care needed for patients undergoing hemodialysis [25]. Otherwise, patients treated at the Kidney Clinic belong to the Ministry of Health and the majority of them are in conditions of poverty and extreme poverty [14]. We also found in this last group a higher percentage of patients are financially dependent on the family, and this could generate frustration and perceive themselves as burdens to their families [26]. This result is supported by various studies in hemodialysis patients with CKD, in which factors such as socioeconomic status and monthly income were associated with HRQoL [27, 28].

Although HRQoL is lower in hemodialysis patients with CKD than in other populations with CKD [2], our findings suggest associations with various factors that contribute to its

reduction, which can serve as a starting point for the design of interventions within the care of this patient group.

## Limitations

Our study had several limitations that should be considered when interpreting our results: Due to the observational nature of our study, we could not establish causality between variables; we only evaluated patients with KF undergoing hemodialysis, and this group was small, making our results not generalizable to the entire Peruvian population. Since self-administered surveys were conducted, there is the potential for social desirability bias, which we attempted to reduce by explaining to patients that the survey would be anonymous; and socioeconomic status was not directly measured but assessed indirectly (educational level and the type of health insurance they have), which may lead to an unclear interpretation of socioeconomic status. In addition, although two researchers had two training sessions in interviews and data collection, information corresponding to the duration of each interview was not collected, which could have helped a better understanding of the data collection process.

## Conclusions

The HRQoL of hemodialysis patients with CKD was higher in the mental than in the physical sphere. The factors associated with a better HRQoL score included being male, having higher education, and financial self-sufficiency. In contrast, a lower HRQoL score was observed in older individuals and those receiving care at the Kidney Clinic. Our findings underscore the imperative of addressing the specific needs of vulnerable populations, recognizing them as potential focal points of healthcare inequities that demand targeted intervention and attention.

## Supporting information

**S1 File. Data collection form.**
(PDF)

**S2 File. Study database.**
(XLSX)

## Author Contributions

**Conceptualization:** Dana Machaca-Choque, Guimel Palomino-Guerra, Javier Flores-Cohaila, Edgar Parihuana-Travezaño, Alvaro Taype-Rondan, Sujey Gomez-Colque, Cesar Copaja-Corzo.

**Data curation:** Dana Machaca-Choque, Guimel Palomino-Guerra.

**Formal analysis:** Cesar Copaja-Corzo.

**Funding acquisition:** Edgar Parihuana-Travezaño, Cesar Copaja-Corzo.

**Methodology:** Alvaro Taype-Rondan, Cesar Copaja-Corzo.

**Writing – original draft:** Javier Flores-Cohaila, Edgar Parihuana-Travezaño, Sujey Gomez-Colque, Cesar Copaja-Corzo.

**Writing – review & editing:** Dana Machaca-Choque, Guimel Palomino-Guerra, Javier Flores-Cohaila, Edgar Parihuana-Travezaño, Alvaro Taype-Rondan, Sujey Gomez-Colque, Cesar Copaja-Corzo.

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
