## [Decision Letter · Decision Letter 0]

6 Feb 2024

PONE-D-23-39698Quality of life and its associated factors in chronic kidney disease patients undergoing hemodialysis from a Peruvian city: a cross-sectional studyPLOS ONE

Dear Dr. Copaja-Corzo,

Thank you for submitting your manuscript to PLOS ONE. After careful consideration, we feel that it has merit but does not fully meet PLOS ONE’s publication criteria as it currently stands. Therefore, we invite you to submit a revised version of the manuscript that addresses the points raised during the review process.

 Please submit your revised manuscript by Mar 22 2024 11:59PM. If you will need more time than this to complete your revisions, please reply to this message or contact the journal office at plosone@plos.org. Please include the following items when submitting your revised manuscript:A rebuttal letter that responds to each point raised by the academic editor and reviewer(s). You should upload this letter as a separate file labeled 'Response to Reviewers'.A marked-up copy of your manuscript that highlights changes made to the original version. You should upload this as a separate file labeled 'Revised Manuscript with Track Changes'.An unmarked version of your revised paper without tracked changes. You should upload this as a separate file labeled 'Manuscript'.If applicable, we recommend that you deposit your laboratory protocols in protocols.io to enhance the reproducibility of your results. Protocols.io assigns your protocol its own identifier (DOI) so that it can be cited independently in the future. For instructions see: https://journals.plos.org/plosone/s/submission-guidelines#loc-laboratory-protocols. Additionally, PLOS ONE offers an option for publishing peer-reviewed Lab Protocol articles, which describe protocols hosted on protocols.io. Read more information on sharing protocols at https://plos.org/protocols?utm_medium=editorial-email&utm_source=authorletters&utm_campaign=protocols.

We look forward to receiving your revised manuscript.

Kind regards,

Chikezie Hart Onwukwe

Academic Editor

PLOS ONE

Journal Requirements:

"We would like to acknowledge the Scientific University of the South for its support in paying the costs of processing the article."

"The author(s) received no specific funding for this work"

4. In the online submission form, you indicated that "The data cannot be shared publicly because they were obtained by request of both the Hemodialysis Unit at the DACH and the Kidney Clinic., who approved access to the data to the authors develop this specific research work. Data is available upon request from the Hemodialysis Unit at the DACH (contact via (052) 583060 , email

ospetacna@essalud.gob.pe and address: Calana Highway Km 6.5, Tacna, Peru) and the Kidney Clinic (contact via (052) 424686 and address: Gregorio Albarracin Street Nro 550, Tacna, Peru) for

researchers who meet the criteria for access to sensitive data."

**Additional Editor Comments:**

Reviewers' comments:

Reviewer's Responses to Questions

**Comments to the Author**

1. Is the manuscript technically sound, and do the data support the conclusions?

Reviewer #1: Partly

Reviewer #2: Yes

2. Has the statistical analysis been performed appropriately and rigorously? 

Reviewer #1: I Don't Know

Reviewer #2: I Don't Know

3. Have the authors made all data underlying the findings in their manuscript fully available?

Reviewer #1: No

Reviewer #2: No

4. Is the manuscript presented in an intelligible fashion and written in standard English?

Reviewer #1: Yes

Reviewer #2: Yes

5. Review Comments to the Author

Reviewer #1: Line 162, only 53.1% had 3 hours session; what about the other 46.9%? Did they have less or more than 3 hours of session? It may be interesting to compare the QOL between those with a higher number of hours per month vs lower numbers of hemodialysis per month.

Reviewer #2: The manuscript describes quality of life amongst dialysis patients in 2 hospitals in the city of Tacna, Peru. Data was collected using telephone surveys with the HrQoL 36 form. The methodology and the exclusion criteria are quite clear.

The authors state that the interviewers had training sessions as follows: "Two researchers (D.M.C. and G.P.G.) underwent two training sessions in interviews and data collection, each lasting 45 minutes". They should also state the average length of the phone interviews for better understanding of the data collection process.

6. PLOS authors have the option to publish the peer review history of their article (what does this mean?). If published, this will include your full peer review and any attached files.

Reviewer #1: **Yes: **Professor Aasim Ahmad

Reviewer #2: **Yes: **Ngozi Virginia Aikpokpo

---

## [Author Response · Author response to Decision Letter 0]

13 Feb 2024

Dear reviewers and editor, in this letter we respond to your observations:

Editor's Comments:

a. It was made according to PLOS ONE's style requirements. 

a. More details were given in the ethics section. Line 145 – 147.

3. Thank you for stating the following in the Acknowledgments Section of your manuscript: "We would like to acknowledge the Scientific University of the South for its support in paying the costs of processing the article." We note that you have provided funding information that is not currently declared in your Funding Statement. However, funding information should not appear in the Acknowledgments section or other areas of your manuscript. We will only publish funding information present in the Funding Statement section of the online submission form. Please remove any funding-related text from the manuscript and let us know how you would like to update your Funding Statement. Currently, your Funding Statement reads as follows: "The author(s) received no specific funding for this work". Please include your amended statements within your cover letter; we will change the online submission form on your behalf.

a. Dear editor, we have made the modification and we have removed the thanks and detailed the APC payment in the corresponding section. 

4. All PLOS journals now require all data underlying the findings described in their manuscript to be freely available to other researchers, either 1. In a public repository, 2. Within the manuscript itself, or 3. Uploaded as supplementary information. This policy applies to all data except where public deposition would breach compliance with the protocol approved by your research ethics board. If your data cannot be made publicly available for ethical or legal reasons (e.g., public availability would compromise patient privacy), please explain your reasons on resubmission and your exemption request will be escalated for approval.

a. Dear editor, we attach the study database as supplementary information (S2_File) in Excel spreadsheet format; Sheet one contains the complete information that was analyzed in the study and sheet two contains the data of each variable as well as its interpretation. 

5. Review your reference list to ensure that it is complete and correct. If you have cited papers that have been retracted, please include the rationale for doing so in the manuscript text, or remove these references and replace them with relevant current references. Any changes to the reference list should be mentioned in the rebuttal letter that accompanies your revised manuscript. If you need to cite a retracted article, indicate the article’s retracted status in the References list and also include a citation and full reference for the retraction notice.

a. We have carried out the corresponding review. 

Revisión:

6. Reviewer #1: Line 162, only 53.1% had 3 hours session; what about the other 46.9%? Did they have less or more than 3 hours of session? It may be interesting to compare the QOL between those with a higher number of hours per month vs lower numbers of hemodialysis per month.

a. Dear reviewer, we appreciate your observation, 46.9% were patients who had hemodialysis lasting 3 and a half hours; we added this information in the manuscript. Although this variable is interesting to analyze, it is the patients treated at EsSalud who underwent hemodialysis for 3 hours and those at the Kidney Clinic who underwent hemodialysis for 3 and a half hours. Therefore, this variable could not be included in the multivariable model. 

7. Reviewer #2: The manuscript describes quality of life amongst dialysis patients in 2 hospitals in the city of Tacna, Peru. Data was collected using telephone surveys with the HrQoL 36 form. The methodology and the exclusion criteria are quite clear. The authors state that the interviewers had training sessions as follows: "Two researchers (D.M.C. and G.P.G.) underwent two training sessions in interviews and data collection, each lasting 45 minutes". They should also state the average length of the phone interviews for better understanding of the data collection process.

a. Your observation about the length of each interview is an interesting point to analyze. But we were not able to obtain this value since we did not take the time of each interview. When conducting the consultation, the interviewers reported that on average the time of each interview was between 20 and 30 minutes. Since this data is not objective, we added it as a limitation of the study. 

We consider that this new version of the manuscript has improved its quality thanks to your comments.

Kind regards

Dr. Cesar Copaja Corzo (Corresponding autor)

---

## [Decision Letter · Decision Letter 1]

26 Feb 2024

Quality of life and its associated factors in chronic kidney disease patients undergoing hemodialysis from a Peruvian city: a cross-sectional study

PONE-D-23-39698R1

Dear Dr.Cesar Copaja-Corzo,

We’re pleased to inform you that your manuscript has been judged scientifically suitable for publication and will be formally accepted for publication once it meets all outstanding technical requirements.

Kind regards,

Chikezie Hart Onwukwe

Academic Editor

PLOS ONE

Additional Editor Comments (optional):

Reviewers' comments:

Reviewer's Responses to Questions

**Comments to the Author**

1. If the authors have adequately addressed your comments raised in a previous round of review and you feel that this manuscript is now acceptable for publication, you may indicate that here to bypass the “Comments to the Author” section, enter your conflict of interest statement in the “Confidential to Editor” section, and submit your "Accept" recommendation.

Reviewer #2: All comments have been addressed

2. Is the manuscript technically sound, and do the data support the conclusions?

Reviewer #2: Yes

3. Has the statistical analysis been performed appropriately and rigorously? 

Reviewer #2: I Don't Know

4. Have the authors made all data underlying the findings in their manuscript fully available?

Reviewer #2: Yes

5. Is the manuscript presented in an intelligible fashion and written in standard English?

Reviewer #2: Yes

6. Review Comments to the Author

Reviewer #2: The authors have address the concerns noted in the previous review. They have stated the limitations of the study, which could prompt other researchers in finding ways to overcome these limitations.

7. PLOS authors have the option to publish the peer review history of their article (what does this mean?). If published, this will include your full peer review and any attached files.

Reviewer #2: **Yes: **Ngozi Virginia Aikpokpo
